# Surgery for Recurrent Pancreatic Cancer: Is It Effective?

**DOI:** 10.3390/cancers11070991

**Published:** 2019-07-16

**Authors:** Lucia Moletta, Simone Serafini, Michele Valmasoni, Elisa Sefora Pierobon, Alberto Ponzoni, Cosimo Sperti

**Affiliations:** 1Department of Surgery, Oncology and Gastroenterology, 3rd Surgical Clinic, University of Padua, 35128 Padua, Italy; 2Department of Radiology, Padua General Hospital, 35128 Padua, Italy

**Keywords:** pancreatic cancer, pancreatic adenocarcinoma, recurrence, redo surgery, completion pancreatectomy

## Abstract

Despite improvements to surgical procedures and novel combinations of drugs for adjuvant and neoadjuvant therapies for pancreatic adenocarcinoma, the recurrence rate after radical surgery is still high. Little is known about the role of surgery in the treatment of isolated recurrences of pancreatic cancer. The aim of this study was to review the current literature dealing with surgery for recurrent pancreatic cancer in order to examine its feasibility and effectiveness. An extensive literature review was conducted according to the 2009 Preferred Reporting Items for Systematic Reviews and Meta-Analyses (PRISMA) guidelines and 14 articles dealing with re-resections for recurrent pancreatic adenocarcinoma were analyzed, focusing on the characteristics of the primary neoplasm and its recurrence, the surgical procedures used, and patient outcomes. Data were retrieved on a total of 301 patients. The interval between surgery for primary pancreatic cancer and the detection of a recurrence ranged from 2 to 120 months. The recurrence was local or regional in 230 patients, and distant in 71. The median overall survival was 68.9 months (range 3–152) after resection of the primary tumor, and 26.0 months (range 0–112) after surgery for recurrent disease. The disease-free interval after the resection of recurrences was 14.2 months (range 4–29). Although data analysis was performed on a heterogeneous and limited number of patients, some of these may benefit from surgery for isolated recurrence of pancreatic adenocarcinoma. Further studies are needed to identify these cases.

## 1. Introduction

Early diagnosis is essential for the successful treatment of pancreatic cancer, and surgical resection is the only potentially curative therapy [1]. Due to the aggressive nature of pancreatic ductal adenocarcinoma (PDAC) and the lack of screening tools for its early diagnosis, most patients present with locally advanced or metastatic disease, making surgical intervention unfeasible [2]. Only 20% of patients are candidates for surgical resection at the time of their diagnosis [3]. The overall 5-year survival rate for PDAC is less than 5%, while for successfully resected patients it ranges from 15% to 40% [4].

Even after radical surgical resection, however, the tumor recurrence rate remains as high as 80%, despite advances in surgical technique and better adjuvant treatments [5]. The disease may relapse in the form of a local, regional or distant recurrence [6]. A local recurrence is usually defined as being isolated to the bed of the pancreatic margin, the pancreatic remnant, or the mesenteric root [7]. Regional recurrences are cancer in the soft tissue or regional lymph nodes outside the confines of the pancreatic bed, or disease in the peritoneal cavity [7]. Distant recurrences involve other organs or lymph nodes [7]. Regardless of the type of relapse, the prognosis is poor, with isolated local recurrences (which account for approximately 30% of patients) have a slightly more favorable prognosis than distant metastatic disease [8,9,10].

For many years there were no established strategies for treating recurrent tumors, and the main options offered to patients were palliative chemotherapy and radiation therapy [11,12,13]. Little is known about the role of surgery for recurrent PDAC, though several studies in recent years have shown a potential benefit of re-resection for PDAC recurrences in selected subgroups of patients.

The aim of this review was to examine the recent English literature to clarify the role of surgery for the treatment of recurrent PDAC.

## 2. Materials and Methods

An extensive literature review was conducted according to the 2009 Preferred Reporting Items for Systematic Reviews and Meta-Analyses (PRISMA) guidelines [14]. A systematic search was performed in PubMed on articles published up to 31 January 2019. Three authors screened all titles and abstracts for eligible articles based on preset eligibility criteria. Full-text manuscripts were examined, and the related references were checked for potentially relevant papers not identified by the initial screening process. The following terms were sought in PubMed: “pancreatic cancer/neoplasm/adenocarcinoma”, “recurrence”, and “surgery/pancreatectomy/redo surgery/completion pancreatectomy”. The related articles function was used to expand the search, and all related abstracts, studies and citations were analyzed. The full text of the selected studies was obtained. Only human studies were considered, and case reports or small series (<5 patients) were disregarded. In the event of successive publications by the same group, only the study with the most comprehensive information on the population examined was included. Articles were excluded from the analysis if patient outcomes were not clearly reported, or it proved impossible to extract the data of interest from the text. The studies included in our review were analyzed in terms of: first author’s name, year of publication, number of patients, details of the primary pancreatic neoplasm, time and site of recurrence, surgical intervention, morbidity, mortality, and long-term outcome. Survival data were recollected patient by patient from each case series when reported by authors.

Statistical analyses were performed in R software (R version 3.6.0—R Foundation for Statistical Computing, Vienna, Austria). Continuous variables are represented as means ± standard deviations or medians with ranges, and categorical variables as percentages or frequencies. Overall and post-recurrence survival curves were constructed with the Kaplan–Meier method.

## 3. Results

After screening titles and abstracts, 243 full-text articles were assessed for eligibility and 14 studies [1,6,15,16,17,18,19,20,21,22,23,24,25,26] were ultimately included in our critical appraisal (Figure 1). Nine papers dealt with isolated local recurrences (ILR) [1,16,17,18,19,20,21,22,26], two with the surgical treatment of lung metastases [24,25], and three reported the surgical results of both local and distant pancreatic recurrences [6,15,23]. Indications for re-resection/exploration in these patients were a strong probability of a potentially-resectable recurrence and no signs of further disseminated disease in patients in acceptable general conditions (ASA, American Society of Anesthesiologists, 1–3). The standard preoperative work-up included contrast-enhanced abdominal computed tomography and endosonography. Additional imaging (magnetic resonance imaging, magnetic cholangiopancreatography, positron emission tomography) was performed when indicated. All patients had previously undergone pancreatic resection for PDAC (R0/R1), and they all had histologically-confirmed carcinoma in the initial specimen.

### 3.1. Patient Demographics and Primary Resection Parameters

Although incomplete, data were available regarding a total of 1281 patients with recurrent PDAC (57.6% males, 42.4% females; mean age 66.25 years); 301 of them underwent surgical treatment for their recurrent disease. However, the real proportion of resected patient over the total number of recurrence, is considerably less. A subgroup analysis of the case series presenting both, the number of patients recurrent and resected [1,6,15,16,17,19,20,21,22,23,25], show a proportion of resection/recurrence rate of 6.4% (82 out of 1281 patients) (Table 1).

The surgical procedures previously performed for the patients’ primary PDAC included 569 pancreaticoduodenectomies (both Whipple and pylorus-preserving), 118 distal pancreatectomies, and 15 total pancreatectomies. Lymph node involvement was reported in 427 cases. Data were available on adjuvant treatments after primary surgery for 370 patients: 358 were given adjuvant chemotherapy, and 12 had radio- and chemotherapy.

The median interval between initial pancreatic surgery and the detection of a recurrence was 24.3 months (range 2–120 months). Chemotherapy or radiation therapy was administered prior to surgery for recurrent disease in three studies [6,18,20]: 21 patients were given chemotherapy; 11 had radio- and chemotherapy; and 11 received external beam radiation therapy.

An ILR or regional recurrence was detected in 230 of the re-resected patients. A distant recurrence occurred in another 71 cases (43 lung metastases, 4 local recurrences associated with a single hepatic metastasis, 15 hepatic metastases with no local recurrence, 3 ovary, 1 brain, 1 abdominal wall implant, 4 multiorgan involvement). 

### 3.2. Surgical Procedures and Postoperative Outcome

Treatment options for ILR included completion pancreatectomy or pancreatic resections (*n* = 202), local tumor bed excision (*n* = 4), and lymph node resection (*n* = 27). An associated procedure was performed in 85 cases to achieve clear margins, involving: 1 celiac axis resection, 12 portal vein resections, 22 other organ resections, and intraoperative radiotherapy (IORT) in 50 cases. Patients with distant metastases were treated with: radiofrequency ablation in 6 cases (associated with liver resection in 2), lung resection in 43, liver resection in 5, hysterectomy and bilateral salpingo-oophorectomy in 1, brain metastasis resection in 1, multiorgan resection in 4, and removal of an abdominal wall implant in 1.

The median intraoperative blood loss was 595 mL (range 110–9865 mL) and the median operating time was 170 min (range 141–1860 min) (Table 2). The median hospital stay was 11 days (range 2–32 days). Postoperative mortality was reportedly low overall, occurring in only 3 cases (1.8% mortality rate). Morbidity data were available in 8/14 studies, and the morbidity rate was 28.1%. The following complications were reported: 9 wound infections, 7 intra-abdominal abscesses, 4 pneumonia, 3 urinary tract infections, 2 perforated gastric ulcers, 2 delayed gastric emptying, 1 pancreatic fistula, 1 biliary leak, 1 burst abdomen, 1 second-degree burns, 1 acute renal failure, 1 deep vein thrombosis, and 1 atrial fibrillation. A R0/R1 resection of the recurrent PDAC was achieved in most of the reported cases (*n* = 132/160, 82.5%), with a limited number of R2 resections (*n* = 19/160, 11.9%).

### 3.3. Survival Analysis

Seventy-eight patients (25.9%) were reportedly alive a median follow-up of 64,5 months (range 3–136.1) after surgical resection for a recurrent PDAC. The median overall survival (OS) was 68.9 months (range 3–152) after resection of the primary tumor, and 26.0 months (range 0–112) after surgery for recurrent disease (Table 3, Figure 2 and Figure 3). The disease-free interval after the resection of a recurrence was 14.2 months (range 4–29).

Six studies [15,18,20,21,23,27] also provided data on 244 patients with recurrences that were not treated surgically due to either locally unresectable disease or disseminated distant disease. In these patients the median overall survival (OS) ranged between 10.8 and 30 months after primary pancreatectomy, and between 6 and 11 months after their recurrent PDAC was diagnosed. In six studies [17,20,22,23,25,26], the median survival was significantly longer for patients who underwent surgery for a recurrence than for unresected patients.

Comparison, between survival curves of resected and unresected patients, was not performed because of the lack of single patient outcome in unresected group.

## 4. Discussion

Recurrences after surgical resection of PDAC are unfortunately still common, the rate being as high as 80%, even after R0 surgery [28,29], with a 27% disease-specific survival at 3 years [30]. PDAC can recur locally (ILR), regionally, or as distant metastases (to the liver, lung or bone) [31]. The median survival after the diagnosis of a recurrence has been found to be as low as 7 months for local recurrences and 3 months for hepatic metastases [8]. For several neoplasms (neuroendocrine tumors, colorectal cancer, renal carcinoma), the surgical resection of recurrent disease has been shown to improve survival, and found to be potentially curative in selected cases [32,33,34]. PDAC recurrences, on the other hand, often involve advanced, disseminated disease, and carry a poor prognosis due to the limited value of the available therapeutic options. When it recurs locally, moreover, PDAC often presents as an unresectable lesion [8]. Some authors found that the early diagnosis of asymptomatic recurrences significantly improves survival by comparison with symptomatic patients, suggesting a possible advantage of detecting recurrences promptly [31]. The pattern of PDAC recurrences varies greatly from one surgical series to another. Sperti et al. [8] reported a local recurrence rate of 72% after primary pancreatectomy, while Griffin et al. [28] found a rate of 19% of ILR. In the Conco-001 trial [34], which investigated the role of adjuvant therapy after curative pancreatectomy, local recurrences were reported in 20% of cases, while distant relapses were more common. In an autopsy series, local relapse was reported in up to 75% of patients [35]. The surgical treatment of recurrent PDAC has historically only been used for palliative purposes, in cases of gastrointestinal or biliary obstruction. Its role remains unclear, given the predictably high associated morbidity and mortality. Judging from our literature review, although second procedures for recurrences appear to be difficult and technically demanding, the related morbidity and mortality rates seem to be acceptable, and no worse than for primary pancreatic resections [36,37]. So repeat resection seems to be feasible and safe in expert hands.

The indications for re-resection in patients with local PDAC recurrences are usually limited to those with ILR and no involvement of the liver, lung, bone or peritoneum. An aggressive surgical approach to recurrent PDAC seems to afford a survival benefit in selected patients, in terms of both overall and disease-free survival, when compared with unresected patients. Strobel et al. [20] reported a median survival of 26 months after re-resection of ILR (*n* = 41) as opposed to 10.8 months in unresected patients (*p* < 0.01). Miyazaki et al. [17] described 11 patients who underwent repeat pancreatectomy for isolated pancreatic remnant recurrences, with a more favorable survival rate than unresected patients (*p* < 0.01). Hashimoto et al. [16] found an OS of 72 months after repeated pancreatectomy in 8 patients with pancreatic remnant recurrences. Thomas et al. [6] reported a 5-year OS rate of 52.4% in patients selected for reoperation, who survived a median 81.1 months, whereas unresected patients had an OS of 18.1 months. Various authors achieved a long-term survival with long-term local control after pancreatic re-resection. Strobel et al. [20] found that 20% of patients who underwent re-resection for ILR survived 5 years or more. Boone et al. [15] had three patients who survived >5 years after re-resection with curative intent. In our analysis, the median OS for patients who underwent resection for primary PDAC was 51 months as opposed to 24 months in unresected patients; and after a second operation for recurrent disease it was 22 months as opposed to 10 in those not treated surgically.

Isolated lung metastases from pancreatic cancer have also been associated with a good outcome after surgical resection. Two different studies [24,25] on the surgical treatment of solitary lung metastases showed a survival benefit compared with solitary distant recurrences at other sites. The indolent nature of lung metastases from PDAC has been suggested by other authors too [38,39], who found that patients with recurrences limited to the lung had a longer median survival after their disease recurred than those whose disease relapsed at other sites. This would suggest a role for surgical resection in cases with a single lung metastasis. Boone et al. [15] also found a trend towards a longer survival for local recurrences (31.8 months) and lung metastases (27.6 months) than for liver metastases (13.9 months): although it did not reach statistical significance, this finding further supports the idea of a possible survival benefit of second surgery for isolated local recurrences and solitary lung metastases.

The liver is the most common site of metastatic PDAC [40]. In colorectal cancer, surgical resection of hepatic metastases has proved oncologically beneficial, positively influencing OS and quality of life [41,42]. Hepatectomy for metastases from neuroendocrine tumors have likewise led to good long-term outcomes and better symptom control [43]. This approach is rarely used for liver metastases from PDAC, however, even at experienced centers [44]. Dünschede et al. [45] reported on 9 cases of metachronous hepatic metastases after primary R0 pancreatic resection: 4 were treated with resection (1 left lateral bisegmentectomy, 3 atypical liver resections), and 5 with chemotherapy with gemcitabine. The median survival was higher in the resection group (31 months; range 20–51 months) than in the chemotherapy group (11 months; range 8–19 months) in the chemotherapy group. Thomas et al. [6] reported six cases of hepatic metastases treated with surgical resection or radiofrequency ablation, achieving a median OS of 32.5 months, while the median OS for patients with locoregional recurrences was 79.3 months, and for those with an isolated lung recurrence it was 92.3 months.

Finally, from literature review analysis, a better survival was obtained after resection of isolated local recurrence in remnant pancreas or lung, while a poorer prognosis was observed after resection of liver and peritoneal metastases. Surgery remains absolutely not indicated in pancreatic recurrence with diffuse perivascular or peritoneal tumor spread [44].

Thomas et al. [6] ran a univariate analysis on the clinical and pathological factors associated with OS in patients with resected PDAC recurrences: the pattern of PDAC recurrence, regional lymph node status at first surgery, and the duration of the first disease-free interval (DFI) were found significantly associated with OS. In particular, patients re-operated for a recurrence with a first DFI longer than 20 months had a longer OS than those with a shorter DFI. Similarly, Boone et al. [15] reported a better survival in patients whose recurrent disease was resected >15 months after a primary resection. These data suggest a role for second surgery in patients with stable isolated recurrences of PDAC.

For many years, radiation therapy and chemotherapy have been the only treatments available for metastatic and recurrent PDAC. In the setting of local recurrences, chemotherapy is often combined with radiation to improve response rates. Wilkowski et al. [46] described a series of 18 patients with ILR treated with radiation plus combination chemotherapy, with a median OS of 17.5 months. Only 37% of patients achieved complete remission. More recently, systemic chemotherapy with FOLFIRINOX (leucovorin calcium or folinic acid, fluorouracil, irinotecan hydrochloride, oxaliplatin) was found to benefit survival in patients with metastatic PDAC and a good performance status by comparison with gemcitabine alone: median progression-free survival and OS were both greatly improved in the FOLFIRINOX group [47]. Another recent phase III clinical trial demonstrated the efficacy of nab-paclitaxel and gemcitabine in improving OS in metastatic PDAC [48].

Adjuvant radiotherapy can be associated with chemotherapy to reduce local recurrence rates. Results in terms of local control are limited, however, due to the therapeutic inadequacy of the radiation doses that can be safely applied to pancreatic tumors [49]. In particular, it is impossible to spare the surrounding small bowel adequately because the close anatomical relationships limit the radiation dose that can be delivered. Stereotactic radiosurgery and intraoperative radiation therapy (IORT) have been proposed as alternative boost techniques for a better local control in advanced pancreatic tumors and recurrent PDAC. The cyberknife system offers a stereotactic boost of radiation, alone or in combination with conventional radiotherapy, with promising results for local control [50]. IORT enables structures at risk to be displaced from the irradiation field during surgery and the target area can be defined and treated under visual control [51]. Despite promising results reported in the literature, however, the role of chemo-radiotherapy—be it as adjuvant therapy after surgical resection of recurrent PDAC or as a solo treatment for relapsing patients—needs to be better clarified by further studies. In particular, more data are needed to distinguish patients who might benefit from an aggressive surgical approach from those eligible for chemotherapy or radiation therapy. The way to combine the available different approaches also needs further investigation.

Our review has some limitations. First of all, the number of patients resected for pancreatic cancer recurrence is small suggesting that only a few patients have isolated recurrence and are thus suitable for surgery (82 out of 1281; 6.4% among all patients who underwent resection for primary pancreatic cancer [1,6,15,16,17,19,20,21,22,23,25]). Despite this is a highly selected group of patients, survival curves suggest a potential benefit from aggressive treatment.

All studies included in this review are retrospective and present a heterogeneous population of patients with local recurrence as well those with metastatic disease, and data may not have been comprehensively collected. The indication for reoperation for selective recurrent pancreatic cancer has not been clearly standardized, and not well established. Moreover, heterogeneity of treatments before and after the recurrence make it difficult to determine the effect of such treatments on clinical outcome; so, selection bias is inevitably. Appropriate inclusion of these patients within prospective international registries should be strongly encouraged in order to obtain a better quality of evidence and recommendation. Further randomized controlled trials and meta analysis are urgently needed to determine efficacy and indication for surgery in recurrent pancreatic cancer according to evidence-based medicine.

## 5. Conclusions

In conclusion, when pancreatic adenocarcinoma recurs after an initial pancreatectomy, various factors need to be carefully considered in order to decide the best treatment approach for the patient. The pattern of any recurrences, the presence of isolated lesions, and the patient’s general condition and preferences all have to be considered to select appropriate candidates for surgical resection. A multidisciplinary and tailored approach is, therefore, recommended in this setting, and further studies are needed to confirm the role of surgery for recurrent PDAC in selected patients.

## Figures and Tables

**Figure 1 cancers-11-00991-f001:**
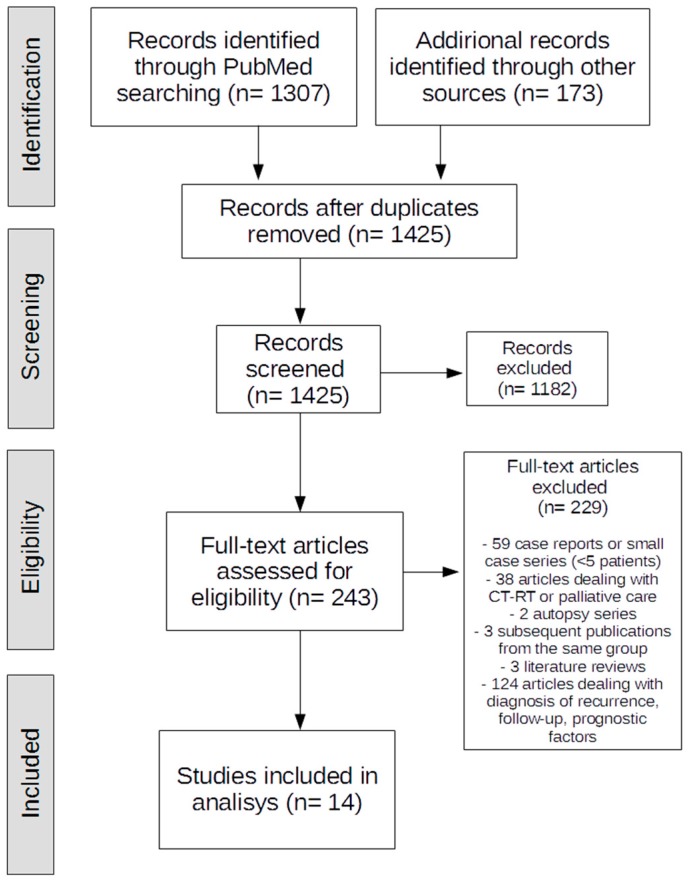
Literature review according to the 2009 Preferred Reporting Items for Systematic Reviews and Meta-Analyses PRISMA guidelines.

**Figure 2 cancers-11-00991-f002:**
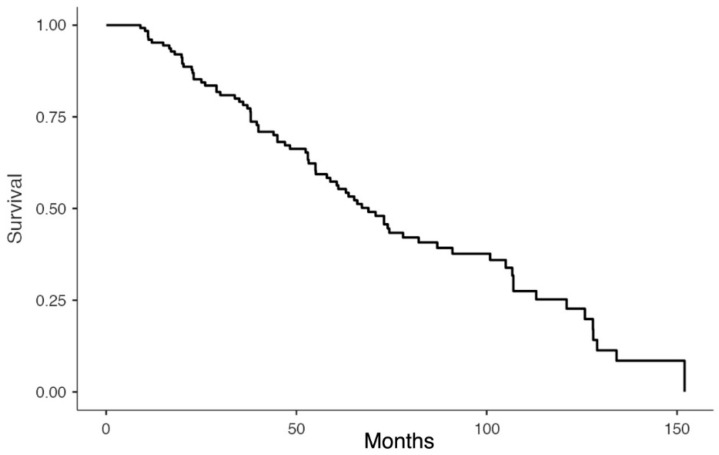
Overall survival after primary pancreatic resection.

**Figure 3 cancers-11-00991-f003:**
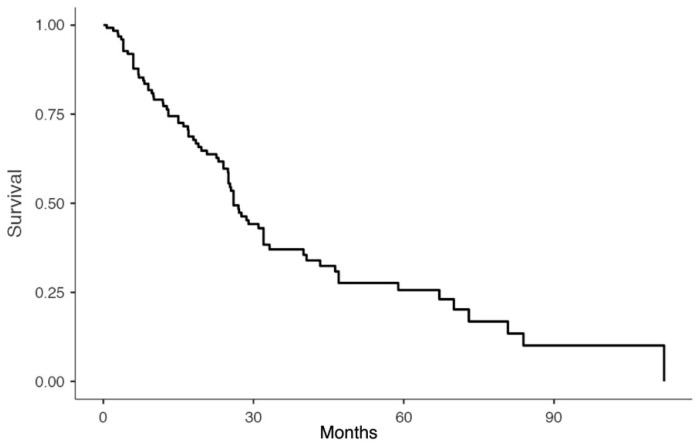
Survival after recurrence resection.

**Table 1 cancers-11-00991-t001:** Characteristics of the primary pancreatic cancer.

Author	Year	N° pts Recurrent/Resected	Age *	Primary T3 *	Primary N+ *	Primary Surgery *			Adjuvant Therapy *
			(Mean; Range)			PD	DP	TP	
Seelig et al. [19]	2009	NR/7	64 (40–76)	11	11	14	2	1	3 RCT, 8 CT
Lavu et al. [1]	2011	NR/11	66 (31–82)	NR	4	7	4	0	6 RCT, 2 CT
Roeder et al. [18]	2012	NR/24	62 (35–75)	29	21	27	5	4	23 CT
Thomas et al. [6]	2012	426/21	62.5 (24.9–84.7)	NR	254	NR	NR	NR	16 CT *, 2 RCT *
Strobel et al. [20]	2013	NR/45	64.1 (55.4–68.4)	NR	NR	74	25	5	NR
Miyazaki et al. [17]	2014	170/11	67 (60–80)	8	4	7	4	0	6 CT*
Boone et al. [15]	2014	NR/22	67 (NR)	13	8	18	4	0	NR
Hashimoto et al. [16]	2014	NR/8	67, 2 (55–80)	4	3	7	3	0	5 CT
Shima et al. [21]	2015	130/6	66,5 (52–82)	4	1	4	2	0	1 CT
Chang et al. [23]	2016	332/14	59 (27–75)	NR	NR	254	NR	NR	114 CT
Yasukawa et al. [24]	2017	NR/12	74,25 (63–83)	NR	NR	6	5	0	11 CT, 1 RCT
Nakayama et al. [22]	2018	127/11	68 (37–73)	NR	8	8	3	0	10 CT
Groot et al. [25]	2018	96/19	68 (60–74)	13	71	77	14	5	84 CT
Yamada et al. [26]	2018	NR/90	64,4 ± 8,3	NR	42	66	47	0	78 CT
Total †		1281/82	66.25 (24.9–84.7)	82	427	569	118	15	358 CT, 12 RCT

Pts: patients; M: male; F: female; N+: lymph node positivity (primary tumor); PD: pancreaticoduodenectomy; DP: distal pancreatectomy; TP: total pancreatectomy; RCT: radio-chemotherapy; CT: chemotherapy; NR: not reported; * data partially avaible; † where individually reported in case series.

**Table 2 cancers-11-00991-t002:** Characteristics of pancreatic ductal adenocarcinoma (PDAC) recurrences.

Author	Time to Recurrence (mo, Range)	Type of Recurrence		Intraoperative Blood Loss (mL, Range)	Operating Time (min, Range)	Hospital Stay (Days, Range)
		Local	Distant			
Seelig et al. [19]	18 (4–120)	7	-	200 (100–800)	255 (150–540)	22 (9–32)
Lavu et al. [1]	32 (2–107)	11	-	500 (200–4000)	NR	6 (5–24)
Roeder et al. [18]	20 (6–76)	24	-	NR	NR	NR
Thomas et al. [6]	22 (2–86)	6	15 (lung 7; liver 6; brain 1, abdominal wall 1)	NR	NR	NR
Strobel et al. [20]	14,9 (11.3–22)	41	4 (local + single liver metastasis)	350 (200–500)	275 (185–345)	9 (7–13)
Miyazaki et al. [17]	32 (7–89)	11	-	867 (310–9865)	231 (160–600)	
Boone et al. [15]	35.1 (7.6–63.6)	10	12 (lung 5, ovary 1, liver 6)	300 (25–6000)	232 (77–435)	8 (2–28)
Hashimoto et al. [16]	32 (23–103)	8	-	NR	NR	NR
Shima et al. [21]	25.1 (12–60)	6	-	465 (110–1390)	165 (141–412)	17 (13–30)
Chang et al. [23]	13.9 (4.4–51.2)	5	9 (liver 3, ovary 2, multiorgan resection 4)	NR	NR	13 (4–29)
Yasukawa et al. [24]	32 (8–74)	0	12 (lung)	NR	NR	NR
Nakayama et al. [22]	24 (10–31)	11	-	NR	NR	NR
Groot et al. [25]	24.3 (9.6–83.0)	0	19 (lung)	NR	NR	3 (2–4)
Yamada et al. [26]	38.6 ± 24.2	90	-	700 ± 618	344 ± 167	27.4 ± 20.0
Total †	28 (2–120)	230	71	595 (110–9865) me	170 (141–1860) me	11 (2–32) me

Mo: months (results reported in median*), IO: intraoperative; mL: milliliter; min: minutes; NR: not reported; me: median; † where individually reported.

**Table 3 cancers-11-00991-t003:** Series reporting treatment and outcome of patients with recurrent PDAC.

Author	Treatment	N° pt	OS (Range)	*p* *	S after R (Range)	*p* °	DFI after R (Range)
Seelig et al. [19]	RESECTED §	7	38 (11–152)	NR	25 (7–41)	NR	NR
	UNRESECTED	10	25 (11–48)		6 (1–29)		NA
Lavu et al. [1]	RESECTED	10	45 (24–114)	NA	13 (1–53)	NA	NR
	UNRESECTED	0	-		-		-
Roeder et al. [18]	RESECTED	18	19 (NR)	NR	NR	NR	17 (7–25)
	UNRESECTED	18	NR		NR		NA
Thomas et al. [6]	RESECTED	20	53.5 (8–128)	NR	7.5 (0.7–112)	NR	9 (1–112)
	UNRESECTED	405	NR		NR		NA
Strobel et al. [20]	RESECTED	41	16.4 (NR)	NR	26 (NR)	<.01	11.4 (NR)
	UNRESECTED	16	10.8 (NR)		10.8 (NR)		NA
Miyazaki et al. [17]	RESECTED	11	78.2 (17–107)	<0.001	25 (3–61)	<.01	24 (3–61)
	UNRESECTED	159	20.3 (NR)		9.3 (NR)		NA
Boone et al. [15]	RESECTED	22	60.6 (11.1–127.9)	NA	28.1 (2.9–80.8)	NA	NR
	UNRESECTED	0	-		-		-
Hashimoto et al. [16]	RESECTED	8	72 (36–129)	NR	17 (10–85)	NR	NR
	UNRESECTED	2	30 (28–32)		10 (9–11)		NA
Shima et al. [21]	RESECTED	6	49 (28–107)	NA	27.5 (6–70)	NA	NR
	UNRESECTED	0	-		-		-
Chang et al. [23]	RESECTED	14	57.8 (NR)	<0.001	14.1 (NR)	<0.001	NR
	UNRESECTED	332	14.0 (NR)		6.9 (NR)		
Yasukawa et al. [24]	RESECTED	12	121 (NR)	NA	47 (6–66)	NA	NR
	UNRESECTED	0	-		-		-
Nakayama et al. [22]	RESECTED	11	70 (NR)	=0.02	19 (3–44)	=0.01	4 (2–25)
	UNRESECTED	35	25 (15–35)		11 (6–25)		NA
Groot et al. [25]	RESECTED	19	68.9 (45.5–92.3)	<0.001	35.0 (21.1–48.9)	=0.002	24.3 (18.8–29.9)
	UNRESECTED #	45	34.2 (23.4–45.1)		20.2 (17.2–23.2)		13.2 (10.4–16.0)
Yamada et al. [26]	RESECTED	90	26 (NR)	=0.012	NR	NR	NR
	UNRESECTED	24	14 (NR)		NR		NR

NR: not reported; NA: not applicable; OS: overall survival; S after R: survival after resection (for patients undergoing surgery), survival after diagnosis of recurrence (for unresected patients); DFI after R: disease-free interval; § Resected: patients undergoing surgery for recurrent disease; Unresected: patients diagnosed with recurrent PDAC but not eligible for resection; * overall survival after primary pancreatectomy for PDAC; ° survival after surgery for recurrence; # 45 patients treated with chemo(radio)therapy.

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
