# Peer review of "Surgery for Recurrent Pancreatic Cancer: Is It Effective?"

_cancers, 2019, doi:10.3390/cancers11070991_

Round 1
Reviewer 1 Report
This is an extensive literature review conducted according to the 2009 PRISMA guidelines evaluating 14 articles dealing with re-resections for recurrent pancreatic adenocarcinoma in order to focus on the characteristics of the primary neoplasm and the recurrence, the surgical procedures used, and patient outcomes. Data were retrieved on a total of 301 patients.
The main conclusion is that some patients may benefit from surgery for isolated recurrent pancreatic adenocarcinoma, but further studies are needed to pinpoint these cases.
The strenghts include:
- Well-structured paper with detailed plan of research
- Strong data on a subject which has not yet been the focus of many preclinical / translational studies
Minor weaknesses: Considering the variability of patients, as stated in the "limitations' the authors, the low number of cases should be reported also in the abstract
Table 2. please revise "Characteristic" with Characteristics"
Author Response
Dear Sir,
We would like to thank the reviewers for the constructive comment to our manuscript “Surgery for Recurrent Pancreatic Cancer: Is It Effective?” (ID: Cancers-546813).
We herewith provide a point-by-point response to the reviewer’s comments and submit the revised manuscript for possible publication in Cancers Special Issue "Advances in Pancreatic Cancer Research".
Reviewer’s comment: Minor weaknesses: Considering the variability of patients, as stated in the "limitations' the authors, the low number of cases should be reported also in the abstractResponse: as the reviewer suggested, we added in the abstract the sentences “Although data analysis was performed on a heterogeneous and limited number of patients, some of these may benefit from surgery for recurrence of isolated pancreatic adenocarcinoma. Further studies are needed to identify these cases.”.
Reviewer’s comment: Table 2. please revise "Characteristic" with Characteristics"
Response: as the reviewer suggested, we have changed Table 2 header.
We thank the reviewer for the positive comment to our manuscript.
Please see the attachment.
Kind Regards,
Prof. Cosimo Sperti
Department of Surgery, Oncology and Gastroenterology,
3rd Surgical Clinic, University of Padua,
Via Giustiniani 2, 35128 Padua, Italy
Phone +390498218845
Fax +390498218821
e-mail: [email protected]

Reviewer 2 Report
The manuscript by Moletta and colleagues analyses surgery for recurrent pancreatic cancer. A systematic review according to PRISMA guidelines was carried out. 14 articles including 301 patients were retrieved. It is shown that the median survival was 68.9 months after resection of the primary tumor, and 26.0 months after surgery for recurrent disease.
This is an interesting and clinically relevant topic with sparse data published so far. The analysis is sound and valid, and the paper is well written. There are some concerns, that should be addressed:
1. “However, individually reported data show a real proportion of resection/recurrence rate of 6.4% (82 out of 1281)”. This should be better explained as it is not readily apparent that this is a subgroup analysis of the studies presenting both, the number of patients recurrent and resected.
2. It would be interesting to discuss different recurrence patterns: recurrence in the pancreatic remnant, locoregional lymph node recurrence, diffuse perivascular/ peritoneal recurrence.
3. It should be further stressed that this is a highly selected group of patients. Only around 6% of patients with suspected local recurrence are resected in the reported series. Survival analyses are also heavily biased.
4. A short discussion whether a clinical trial is feasible should be included. How can we obtain better evidence whether surgery for recurrent pancreatic cancer is helpful, e.g. prospective international registries?
Author Response
Dear Sir,
We would like to thank the reviewers for the constructive comment to our manuscript “Surgery for Recurrent Pancreatic Cancer: Is It Effective?” (ID: Cancers-546813).
We herewith provide a point-by-point response to the reviewer’s comments and submit the revised manuscript for possible publication in Cancers Special Issue "Advances in Pancreatic Cancer Research".
Reviewer’s comment 1: “However, individually reported data show a real proportion of resection/recurrence rate of 6.4% (82 out of 1281)”. This should be better explained as it is not readily apparent that this is a subgroup analysis of the studies presenting both, the number of patients recurrent and resected.
Response: as the reviewer suggested, we added in the paper the following sentences: “However, the real proportion of resected patient over the total number of recurrence, is considerably less. A subgroup analysis of the case series presenting both, the number of patients recurrent and resected [1,6,15,16,18,20-24,26], show a proportion of resection/recurrence rate of 6.4% (82 out of 1281 patients).”.
Reviewer’s comment 2: It would be interesting to discuss different recurrence patterns: recurrence in the pancreatic remnant, locoregional lymph node recurrence, diffuse perivascular/ peritoneal recurrence.
Response: We have already discussed different recurrent patterns (isolated local recurrence, remnant pancreas, lung and liver isolated metastases, peritoneal isolated recurrence). However, as the reviewer suggested, we added in the paper the following sentences: “Finally, from literature review analysis, a better survival was obtained after resection of isolated local recurrence in remnant pancreas or lung, while a poorer prognosis was observed after resection of liver and peritoneal metastases. Surgery remains absolutely not indicated in pancreatic recurrence with diffuse perivascular or peritoneal tumor spread [44].”
Reviewer’s comment 3: It should be further stressed that this is a highly selected group of patients. Only around 6% of patients with suspected local recurrence are resected in the reported series. Survival analyses are also heavily biased.
Response: We have already discussed selection bias of patients for survival analysis. However, as the reviewer suggested, we added in the paper the following sentence: “Despite this is a highly selected group of patients, survival curves suggest a potential benefit from aggressive treatment.”.
Reviewer’s comment 4: A short discussion whether a clinical trial is feasible should be included. How can we obtain better evidence whether surgery for recurrent pancreatic cancer is helpful, e.g. prospective international registries?
Response: as the reviewer suggested, we added in the paper the following sentences: “Appropriate inclusion of these patients within prospective international registries should be strongly encouraged in order to obtain a better quality of evidence and recommendation. Further randomized controlled trials and meta analysis are urgently needed to determine efficacy and indication for surgery in recurrent pancreatic cancer according to evidence based medicine.”.
We thank the reviewer for the positive comment to our manuscript.
Please see the attachment.
Kind Regards,
Prof. Cosimo Sperti
Department of Surgery, Oncology and Gastroenterology,
3rd Surgical Clinic, University of Padua,
Via Giustiniani 2, 35128 Padua, Italy
Phone +390498218845
Fax +390498218821
e-mail: [email protected]
